# Nutrition and Risk of Stroke

**DOI:** 10.3390/nu11030647

**Published:** 2019-03-17

**Authors:** J. David Spence

**Affiliations:** Divisions of Neurology and Clinical Pharmacology, and Director, Stroke Prevention & Atherosclerosis Research Centre, Robarts Research Institute, Western University, 1400 Western Road, London, ON N6G 2V4, Canada; dspence@robarts.ca; Tel.: +1-519-931-5731

**Keywords:** stroke, nutrition, Mediterranean diet, B vitamins, homocysteine, dietary cholesterol, eggs, sodium

## Abstract

Nutrition is far more important in stroke risk than most physcians suppose. Healthy lifestyle choices reduce the risk of stroke by ~80%, and of the factors that increase the risk of stroke, the worst is diet: only ~0.1% of Americans consume a healthy diet, and only 8.3% consume a somewhat healthy diet. The situation is probably not much better in most other countries. A Cretan Mediterranean diet, high in olive oil, whole grains, fruits, vegetables and legumes, and low in cholesterol and saturated fat, can reduce stroke by 40% or more in high-risk patients. The role of the intestinal microbiome in cardiovascular risk is emerging; high levels of toxic metabolites produced by intestinal bacteria from meat (particularly red meat) and egg yolk are renally excreted. Patients with renal impairment, including the elderly, should limit red meat and avoid egg yolk, as should other patients at high risk of stroke. Salt intake should be limited to 2–3 grams per day. Metabolic B12 deficiency is common and usually missed. It has serious neurological consequences, including an increase in the risk of stroke. It now clear that B vitamins to lower homocysteine reduce the risk of stroke, but we should probably be using methylcobalamin instead of cyanocobalamin.

## 1. Introduction and Background

Most physicians markedly underestimate the importance of lifestyle. In the US Health Professionals Study and the Nurses Health Study, healthy lifestyle choices were defined as not smoking, a moderate consumption of alcohol, maintaining a healthy weight (a body mass index (BMI) < 25), 30 min of daily exercise, and a healthy diet score in the top 40%. Among 43,685 men and 71,243 women in the US, unhealthy behaviours accounted for half of stroke risk; participants who followed all five healthy lifestyle choices had an 80% reduction of stroke [1]. Following all five healthy lifestyle choices also reduced coronary artery disease by more than 80% among Swedish men with hypertension and hyperlipidemia [2]. In the US, and probably also in other developed countries, the most prevalent risk factor is an unhealthy diet: the 2015 statistical report of the American Heart Association reported that only 0.1% of Americans consume a healthy diet, and only 8.3% consume a somewhat healthy diet (Figure 1).

Diet is not only a major problem in high-income countries. In China, with increasing prosperity, the consumption of meat and eggs has increased markedly in the past 20 years, and there has been a decrease in consumption of fruits and vegetables. This change has been associated with a 26.6% increase in stroke mortality and a 213% increase in coronary mortality, between 2003 and 2013 [3].

## 2. Issues to Be Discussed

In this narrative review I discuss diet, maintenance of a healthy weight, dietary cholesterol, the intestinal microbiome, consumption of egg yolk and red meat, metabolic B12 deficiency, B vitamins for stroke prevention, and salt intake. Other vitamins and supplements will not be discussed; a recent meta-analysis [4] reported that only B vitamins reduced the risk of stroke.

### 2.1. Diet

As shown above, diet is the worst of the health issues in the United States; this is probably also true for most high-income countries. There are several issues to consider in regard to diet: what is most important is probably the eating pattern, rather than any particular food.

#### 2.1.1. Mediterranean Diet

It was discovered in the Seven Countries Study that in Crete, the coronary risk was 1/15th that in Finland, and only 40% of that in Japan. In a retrospective article, Ancel Keys, the leader of that study, described it as a “mainly vegetarian diet, favouring fruit for dessert instead of pastries … much lower in meat and dairy. The diet is high in olive oil, whole grains, fruits, vegetables, and legumes, and low in cholesterol and animal fat. The low-fat diet recommended for many years for reduction of coronary risk was, as described by Willett and Stampfer [5], “pulled out of thin air”, by a committee imagining a diet that would lower fasting cholesterol levels. However, as shown in Figure 2, it is the kind of fat that matters. In Crete 40% of calories were from fat, but it was mainly olive oil. In Finland 38% of calories were from fat, but mainly animal fat, accompanied by its evil companion cholesterol. In Japan, only 10% of calories were from fat, largely from fish.

In recent years there has been a backlash against the low-fat diet, with the recognition that a diet high in carbohydrates is probably harmful. However, the right answer to avoiding a high-carb diet is not to consume diets that are high in animal fat, such as the “keto diet” or the Atkins diet: it is to consume the Mediterranean diet, which, being high in calories from fat, is a low-glycemic index diet. Perhaps the best illustration of this was the Israeli diet study [6] in which overweight residents of a nuclear facility were randomized to a low-fat diet, a low-carb diet similar to the Atkins diet, or a Mediterranean diet. In that study, remarkable adherence was obtained because meals were taken in a cafeteria, a different dietician championed each diet, and meals in the cafeteria were colour-coded so that people would not take the wrong food. Adherence was 95% at a year and 86% at two years, much better than in other studies in free-living human beings. Weight loss was identical on the low-carb and Mediterranean diet, and significantly better than on the low-fat diet. As or more importantly, fasting glucose, fasting insulin and insulin resistance were clearly the best on the Mediterranean diet [6].

The first randomized trial, the Lyon Diet Heart Study, in survivors of myocardial infarction, compared the Mediterranean diet (with canola margarine substituted for butter) to a “prudent Western diet” that was lower in fat than the usual diet in France. The Mediterranean diet was associated with 70% reduction of stroke and myocardial infarction in 4 years [7]. That was more than twice the effect of simvastatin in the contemporaneous Scandinavian Simvastatin Survival Study, also conducted in survivors of myocardial infarction. In that study, simvastatin reduced recurrent myocardial infarction by 40% in 6 years, compared to placebo [8].

More recently, in high-risk primary prevention, a diet aiming for low fat intake was compared with two versions of a Mediterranean diet in a Spanish study. One version of the Mediterranean diet was supplemented with olive oil, the other supplemented with mixed nuts. In five years, the nut-supplemented diet reduced stroke by 46% and the olive oil supplemented version reduced stroke by 35% [9]. The study was criticized for methodological problems (persons in the same household were assigned to the same diet after one was randomized, and at one study site all participants were assigned to a Mediterranean diet); but a re-analysis confirmed the results of the initial publication [9].

Little is known about the effect of diet on stroke subtypes. Chen et al. [10] reported from a network meta-analysis that the Mediterranean diet was associated with a reduction of both ischemic stroke and hemorrhagic stroke. The relative risk for ischemic stroke was 0.86, 95% CI 0.81–0.91; for hemorrhagic stroke it was 0.83, 95% CI 0.74–0.93.

#### 2.1.2. Maintenance of a Healthy Weight

Obesity is associated with socioeconomic inequality in low and middle-income countries [11], and it is an established risk factor for stroke incidence [12] and mortality [13]. Each 5 kg/m^2^ increase in BMI, within the range of 25–50 kg/m^2^ is associated with about 40% higher stroke mortality [13].

For the most part, maintaining a healthy weight requires limiting caloric intake; for people who are mostly sedentary, weight is maintained with approximately 10 kilocalories (calories) per pound per day. Most patients vastly overestimate the number of calories burned by various activities; a recent development is the concept of exercise “calorie equivalents”. To burn off 250 calories from a small burger in a fast food restaurant would require walking 2.6 miles [14]; i.e. only 96 calories are burned per mile. One pound of body fat contains ~3500 stored calories, so to lose a pound a person must cut out ~3500 calories or walk about 36 miles. Caloric restriction prolongs life in animal models, perhaps by reducing oxidative stress; it is thought that the reasons that Okinawa has the highest proportion of centenarians are that the diet resembles the Mediterranean diet [15], and caloric restriction is the norm in that district [16].

Exercise has other benefits for stroke prevention [17], and probably reduces insulin resistance, but most obese patients will not have a substantial weight loss without caloric restriction. Besides genetic contributions to obesity, there may also be effects of the intestinal microbiome on the proportion of consumed calories that are absorbed. Some patients may benefit from bariatric surgery if they are unable to lose weight by adopting a lower-calorie diet.

### 2.2. Dietary Cholesterol

The common statement that dietary cholesterol does not raise fasting levels of low density lipoprotein cholesterol is a red herring, promulgated by the egg industry and the meat industry [18]. Diet is not about the fasting state; it is about the post-prandial state [19]. In fact dietary cholesterol does raise fasting lipids, with important individual differences. However, as dietary cholesterol is usually consumed with saturated fat, it is crucial to recognize that cholesterol has a permissive effect on the harmful effects of saturated fat [20]. For ~4 h after a high-fat/high cholesterol meal, there is endothelial dysfunction, oxidative stress, and arterial inflammation [21,22].

What matters is not the effect of dietary cholesterol on risk factors, but the effect of dietary cholesterol on cardiovascular disease. In animal models dietary cholesterol causes atherosclerosis, and in human beings, dietary cholesterol increases cardiovascular risk [23,24]. There are good reasons for longstanding recommendations that persons at risk of vascular disease consume less than 200 mg/day of cholesterol [22]. In the recent US guideline, it was unfortunate that the first two paragraphs of the press releases were not reversed. The first said that there was insufficient evidence on which to recommend a particular number of milligrams of cholesterol should be consumed, so earlier recommendations to limit cholesterol intake to 300 mg/day for healthy persons would be removed from the guideline. This statement was spun by the egg and meat industries into headlines that essentially said “It’s OK to eat cholesterol now; the new guideline says so.” Not noticed was the second paragraph of the press release, which said “However, the intake of cholesterol should be a low as possible within the recommended eating pattern”.

What is seldom considered is the meaning of “at risk for cardiovascular disease”. The lifetime risk of cardiovascular disease is very high for persons living in high-income countries: approximately 60% by age 85 [25]. This means that the only people who can consume a high cholesterol intake with impunity are those who know they will die or be killed at a young age.

### 2.3. Intestinal Microbiome

An important recent development is the recognition that the intestinal microbiome interacts importantly with diet to increase cardiovascular risk [26]. Toxic metabolic products of the intestinal microbiome, produced mainly from dietary protein, carnitine (largely from red meat) [27] and phosphatidylcholine (largely from egg yolk) cause atherosclerosis in animal models [28], and increase cardiovascular risk. They are mainly excreted by the kidneys, so persons with severe renal failure have very high plasma levels of some of these metabolites (50 to 100-fold higher than in persons with normal renal function). Carnitine and phosphatidylcholine are converted by the intestinal bacteria to trimethylamine, the compound that accounts for the fishy breath of uremic patients [29]. Trimethylamine is in turn oxidized by the liver to trimethylamine N-oxide (TMAO), which causes atherosclerosis in animal models. Among patients referred to the Cleveland Clinic for coronary angiograms, plasma levels in the highest quartile (obtained after a test dose of two hard-boiled eggs) increased the 3-year risk of stroke, myocardial infarction or vascular death 2.5-fold [30]. In patients with impaired renal function, high levels of TMAO accelerate the decline of renal function and increase mortality [31]. As these toxins are renally excreted, they might be termed gut-derived uremic toxins (GDUT).

Our group measured plasma levels of seven GDUT in a study of extremes of carotid atherosclerosis: TMAO, p-cresyl sulfate, hippuric acid, indoxyl sulfate, p-cresyl glucuronide, phenyl sulfate and phenylacetylglutamine [32]. Plasma levels of four of these metabolites (TMAO, p-cresyl sulfate, p-cresyl glucuronide, and phenylacetylglutamine) were found to be significantly higher among patients with severe atherosclerosis not explained by traditional risk factors (“Unexplained Atherosclerosis”), and significantly lower among patients with normal carotid arteries despite high levels of traditional risk factors (a “Protected” phenotype) [32]. This was not explained by differences in renal function, nor by differences in dietary intake of the nutritional precursors of the GDUT. In linear regression, TMAO and p-cresylsulfate were significant predictors of carotid plaque burden, in a model that included a broad panel of traditional risk factors.

Although in the past it was thought that high levels of the GDUT occurred mainly in persons with severe renal failure, we reported that plasma levels of all seven GDUT were significantly elevated with even moderate renal impairment (an estimated glomerular filtration rate (eGFR) in the lowest quartile, (<66 mL/min per 1.73 m^2^). We previously reported that among patients above age 80 the mean eGFR was <60 mL/min per 1.73 m^2^) [33], so persons with impaired renal function, including the elderly, should limit their intake of meat (particularly red meat), and as discussed below, should avoid egg yolk.

#### Consumption of Egg Yolk

Egg consumption increases plasma levels in a dose-dependent manner [34]. One large (65g) egg yolk contains 237 mg of cholesterol, and ~250 mg of phosphatidylcholine. A dietary monstrosity, the Hardee’s Monster Thickburger (a 12-ounce beef burger with bacon and cheese) contains 265 mg of cholesterol [35] and ~350 mg of carnitine. (https://www.hardees.com/menu/nutritional_calculator) Thus one egg yolk is nearly as bad as a monster thickburger, and two egg yolks contain more cholesterol and more TMAO precursor than the monster burger. As the monster burger amounts to more than 4 days’ worth of meat on the mainly vegetarian Mediterranean diet, this means that persons at risk of cardiovascular disease (i.e., most persons in high-income countries) should never eat egg yolks: even one egg a week (52 a year) would constitute nearly 200 extra days’ worth of cholesterol and TMAO precursor.

Egg white is a good source of protein. I recommend to my patients that they learn to make tasty omelets, frittatas and egg salad sandwiches using egg white-based substitutes such as Egg Creations, Egg Beaters and Better ‘N Eggs (another alternative would be tofu-based egg substitutes.) This is an easy way for patients to get a meatless day. I recommend to patients that they go meatless three days a week, and give them recipes for other meatless days, such as chilis, pastas, stir-fries, curries, and a list of internet sites where they can find many more recipes (the recipes can be downloaded from http://www.imaging.robarts.ca/SPARC/). Many patients think that chicken and fish are not “meat”, so they have to be told that vegetarian means nothing all day with eyes, a face or a mother.

The widespread belief that egg yolks are harmless is almost entirely the result of egg industry propaganda. The issue of propaganda of the meat and egg industry, and the food industry in general, was reviewed in 2017 [18]. Table 1 lists some online videos posted by Dr. Michael Greger, explaining the basis of egg industry propaganda.

The two pillars of the egg industry propaganda are the red herring described above (that eggs don’t raise fasting levels of LDL-C by much), and a half-truth. The half-truth is the oft-repeated statement that “eggs can be part of a healthy diet for healthy people”. That statement is based on two US studies, that reported that an egg a day “only” doubled coronary risk among persons who became diabetic during followup [36,37]. However, as shown in Figure 1 above, the US diet is so bad that it is hard to show that anything makes it worse. That principle was discussed by Rose [38]. In Greece, however, where the Mediterranean diet is the norm, the harm from eggs is more obvious: among Greek diabetics, an egg a day was associated with a 5-fold increase in coronary risk, and even 10 grams a day of egg increased coronary risk by 59% [39].

It does not matter that eggs do not increase fasting LDL-C by much. What matters is that they increase coronary risk. They also increase the risk of diabetes [40,41]. Stopping egg intake after a myocardial infarction or stroke would be like quitting smoking after the lung cancer is diagnosed.

### 2.4. Salt Intake

Notwithstanding efforts of the salt industry to obfuscate the issues, it is clear that high salt intake increases the risk and severity of hypertension. High salt intake also leads to depletion of potassium and magnesium, as potassium and magnesium are excreted by the renal tubule in exchange for sodium. An optimal salt intake is ~2–3 g daily, and in most populations the average salt intake is much higher. People who add salt to their food before tasting it are probably consuming ~10 grams per day, and those who add salt after tasting are consuming around 5 grams per day. The body only requires ~0.5 grams per day (except in very hot conditions), so people who add salt before tasting their food are consuming ~20 times as much salt as they need. In order to reduce salt intake to 2–3 grams per day it is necessary not only to avoid added salt, but to read labels and avoid salty food. A dill pickle contains ~2 g of salt, and a cup of canned soup contains ~ a gram; cured meats and bread are also high in salt.

This is a particular problem in China, where the average salt intake is ~7 grams per day, but is much higher in the Northwest. The relationship of sodium intake to hypertension in China was reviewed in 2014 [42]. Most of the sodium (~75%) in the diet is from soy sauce and added condiments.

Restriction of sodium intake has the potential to improve blood pressure control, particularly in patients with higher blood pressures [43], at little/no financial cost. Salt restriction is not as difficult as most patients expect it to be. Adding salt to food causes receptor downregulation of the salt taste buds on the tongue, so patients who regularly add salt cannot taste the salt already in the food, and have to add some. Cutting out salt, and replacing it with other choices to flavour their food, such as lemon juice, vinegar, ginger, spices, herbs and hot peppers, permit up-regulation of the salt taste receptors and assist with sustained salt restriction.

### 2.5. Metabolic B12 Deficiency

Most physicians tend to assume that if the serum B12 is “normal”, there is no problem. However, serum levels of vitamin B12 represent total cobalamin, of which only ~6%–20% is active. In order to assess functional B12 it is necessary to measure holotranscobalamin (seldom measured in North America), or one of the metabolites that become elevated in B12 deficiency: methylmalonic acid, or plasma total homocysteine (tHcy). In the lower half of the distribution of serum B12 (below ~258 pmol/L), approximately 30% of patients have metabolic B12 deficiency, with hyperhomocysteinemia. Hyperhomocysteinemia is a clotting factor, increasing the risk of retinal vein thrombosis, cerebral sinus thrombosis, and deep vein thrombosis. It also seems to increase the risk of platelet aggregates. Hyperhomocysteinemia increases the risk of stroke; in particular it increases the risk of stroke in atrial fibrillation ~4-fold. Most physicians tend to assume that if the serum B12 is “normal”, there is no problem. However, the inflection point of serum B12 at which plasma levels of methylmalonic acid and tHcy become elevated is ~400 pmol/L. To be confident that active B12 levels are adequate, it is necessary for the serum B12 to be ~400 pmol/L (Figure 3).

In 2006, Metabolic B12 deficiency was present among patients with stroke/transient ischemic attack in 10% of patients below age 50 and in 30% of those age ≥ 71 [45]. In 2009, a tHcy > 14 was present in 40% of stroke/TIA patients above age 80 [46]. Although in recent years there appears to have been more B12 supplementation in our region (biochemical B12 deficiency, a serum B12 <156 pmol/L has declined from 10% to 5% of patients since 2009), metabolic B12 deficiency and hyperhomocysteinemia remain common; above age 80, 15% of patients had metabolic B12 deficiency and 35% had hyperhomocysteinemia [47]. As these conditions are common, have serious consequences and are easily treatable (see below), serum B12 and tHcy should be assessed in all patients with stroke or TIA.

### 2.6. B Vitamins for Stroke Prevention

It was clear in the 1980s that high levels of tHcy were a strong, graded independent risk factor for cardiovascular risk. B vitamins are key co-factors in the metabolism of tHcy, so beginning in 1994, randomized controlled trials were conducted to determine if B vitamin supplementation could reduce the risk of stroke. Figure 4 shows the role of B vitamins in metabolism of tHcy.

The first major trial, the Vitamin Intervention for Stroke Prevention (VISP) trial, published in 2004, showed no benefit of combinations of cyanocobalamin (B12), folic acid and pyridoxine (B6) [48]. However, there were three important problems with VISP. 1. We did not have a placebo comparator; hoping to minimize non-compliance, we compared low-dose B vitamins with higher-dose B vitamins; the low-dose vitamin contained 6 mcg of B12, the recommended daily allowance at the time. 2. Folic acid fortification of the grain supply in North America coincided with the initiation of the trial, thus minimizing the benefit of folic acid, and 3. Participants with a low serum B12 at baseline received B12 injections regardless of their randomized assignment, thus negating the benefit of B12 in the very patients who would have benefited most.

Then in 2006, in the Norwegian Vitamin Study there was a slight but significant risk in the arm of the study that included cyanocobalamin [49]. In the same issue of the *New England Journal of Medicine*, the HOPE-2 trial [50] reported a significant 23% reduction of stroke with B vitamins, but the authors, being cardiologists and therefore innocent of the cerebral circulation, concluded that since there was no reduction of myocardial infarction, the reduction of stroke was chance finding. In the same issue of the journal, Loscalzo hypothesized B vitamins were not beneficial because of high levels of unmetabolized folate [51].

In 2005, a subgroup analysis of the VISP trial reported that among participants with good renal function (excluding those in the lowest 10% of eGFR), and excluding those who received B12 injections, there was a 34% reduction of stroke/MI/vascular death among those who entered the study with a serum B12 above the median (meaning they could absorb B12 well) and received high-dose B vitamins, compare with those who had a low baseline serum B12 and received low-dose B vitamins.

Then in 2010, there were two publications that put the cat among the pigeons. The French trial of folic acid and omega-3 fatty acids [52] reported a 43% reduction of stroke with B vitamins with a much lower dose of cyanocobalamin, and a trial in patients with diabetic nephropathy reported harm from high-dose B vitamins including 1 mg per day of cyanocobalamin: there was an acceleration of the decline in renal function, and a doubling of cardiovascular events [53]. In 2011, Spence and Stampfer hypothesized that the reason the early trials were negative was that harm from cyanocobalamin among participants with impaired renal function cancelled out the benefit among participants with good renal function [54].

Then in 2015, the light went on: the China Stroke Primary Prevention Trial (CSPPT) reported that in primary prevention, folic acid reduced stroke by 25% [55]; in higher risk groups the benefit was greater [56]; in patients with tHcy > 15 and low platelet counts the reduction of stroke was 70% [57]. The key finding, for purposes of understanding the issues, was that among participants with impaired renal function, folic acid slowed the decline of renal function, and reduced a composite outcome that included mortality [58].

In 2017, a meta-analysis stratified by renal function and dose of cyanocobalamin revealed that it was probably harm from cyanocobalamin among participants with renal impairment that had obscured the benefit of B vitamins in stroke prevention [59]. A subsequent meta-analysis confirmed that folic acid and B vitamin combinations reduce the risk of stroke [4]. However, we should be using methylcobalamin or oxocobalamin instead of cyanocobalamin [59,60].

### 2.7. Other Vitamins and Supplements Including Omega 3 Fatty Acids

The issue of supplements and vitamins for cardiovascular prevention was extensively reviewed, and meta-analysis performed, in 2018 [4]. Only folic acid and B vitamins reduced the risk of stroke. The authors commented that at the time of the meta-analysis, which found no benefit from omega-3 fatty acids, a study was under way with a higher dose of marine omega-3 fatty acids (2000 units). That trial, published in 2019, randomized 25,871 participants (men aged >50 and women aged ≥55) to omega 3 fatty acids vs. placebo in prevention of cardiovascular disease and cancer [61]. There was no significant reduction of major cardiovascular events or cancer with omega-3 fatty acids. Total myocardial infarction was reduced significantly (hazard ratio (HR) 0.72 (95% confidence interval (CI), 0.59 to 0.90). With regard to total stroke, the HR was 1.04 (95% CI, 0.83 to 1.31; for ischemic stroke the HR was 0.96 (0.74–1.24) [61]. There is some evidence that fish consumption is associated with a reduction of stroke risk [62]. While intake of marine omega-3 fatty acids may account for much of the benefit, Yamori et al. [63] have suggested that taurine intake from fish may also be important.

In a Danish cohort study [64], 57,053 participants aged 50 to 65 years at enrolment were followed for 13.5 years. During followup there were 1879 cases of ischemic stroke. Those cases and a random sample of 3203 subjects from the whole cohort had their fatty acid composition of adipose tissue determined by gas chromatography. Adipose tissue concentration of eicosopentanoic acid was associated with a reduction of total ischemic stroke (HR, 0.74; 95% CI, 0.62–0.88) when comparing the highest with the lowest quartile. “Also, lower rates of large artery atherosclerosis were seen with higher intakes of total marine n-3 polyunsaturated fatty acids,(HR, 0.69; 95% CI, 0.50–0.95), eicosopentanoic acid (HR, 0.66; 95% CI, 0.48–0.91) and docosahexanoic acid (HR, 0.72; 95% CI, 0.53–0.99), and higher adipose tissue content of EPA (HR, 0.52; 95% CI, 0.36-0.76). Higher rates of cardioembolism were seen with higher intakes of total marine n-3 PUFA (HR, 2.50; 95% CI, 1.38–4.53) and DHA (HR, 2.12; 95% CI, 1.21–3.69) as well as with higher adipose tissue content of total marine *n*-3 PUFA (HR, 2.63; 95% CI, 1.33–5.19) and DHA (HR, 2.00; 95% CI, 1.04–3.84). The EPA content in adipose tissue was inversely associated with small-vessel occlusion (HR, 0.69; 95% CI, 0.55–0.88).”

## 3. Discussion

There have been important advances in the understanding of nutrition in stroke prevention. The benefits of a Mediterranean diet, the role of the intestinal microbiome, the harm from egg yolk and the benefits of B vitamins for stroke prevention are all important advances. Further research is needed to establish whether methylcobalamin reduces the risk of stroke.

## 4. Conclusions

Patients at risk of stroke should keep their salt intake to 2–3 grams per day, learn to make a Mediterranean diet tasty and enjoyable, should limit their intake of meat (particularly red meat) and should avoid egg yolk. Such changes represent a challenge for many patients, so physicians should educate patients on the importance of these measures, and provide them with assistance. They should also have metabolic B12 deficiency and hyperhomocysteinemia detected and treated, probably with methylcobalamin. Such measures have the potential to greatly reduce the risk of stroke, so deserve greater attention from the public, and particularly from physicians.

## Figures and Tables

**Figure 1 nutrients-11-00647-f001:**
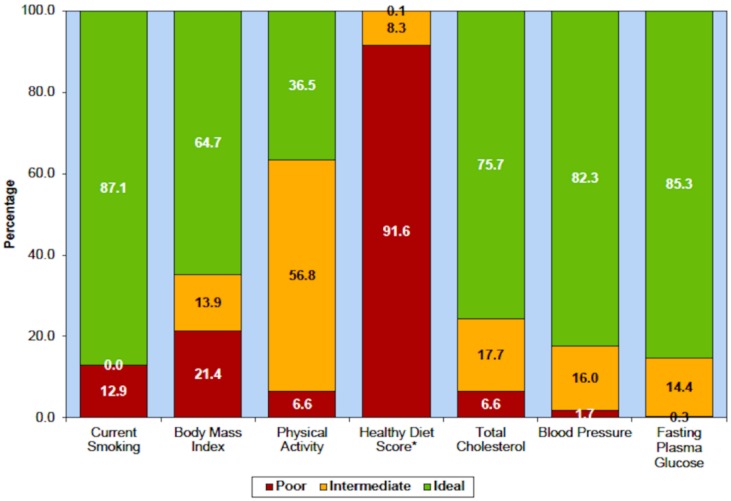
Diet is the worst of the risk issues in the USA. Prevalence (unadjusted) estimates for poor, intermediate and ideal cardiovascular health for each of the seven metrics of cardiovascular health in the American Heart Association 2020 goals, US children aged 12–19 years, National Health and Nutrition Examination Survey (NHANES) 2011–2012. * Healthy diet score data reflect 2009–2010 NHANES data. (Reproduced by permission of Wolters Kluwer from: Mozaffarian D, Benjamin EJ, Go AS, Arnett DK, Blaha MJ, Cushman M, et al. Heart disease and stroke statistics—2015 update: A report from the American Heart Association. Circulation. 2015;131(4): e29–322.).

**Figure 2 nutrients-11-00647-f002:**
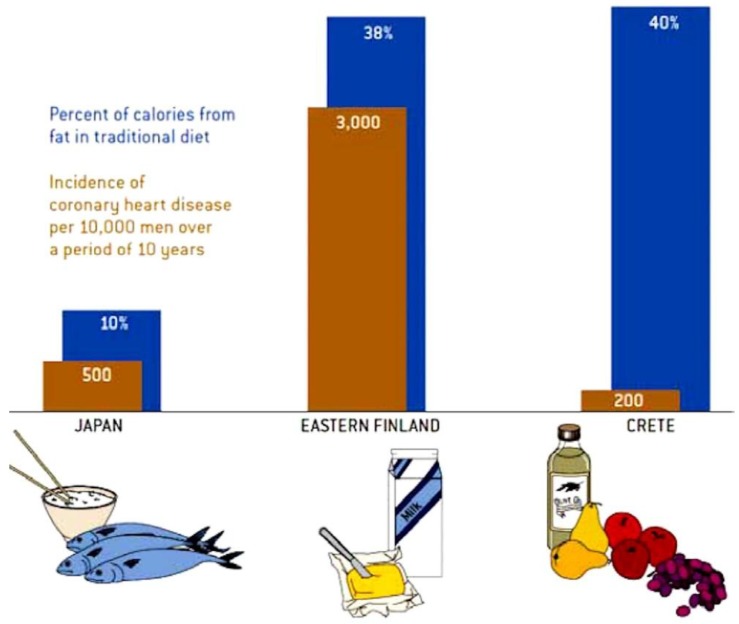
The Mediterranean diet. The Mediterranean diet is a high fat/low glycaemic diet with 40% of calories from fat; however, the fat is mainly beneficial oils such as olive and canola. Among men in the Seven Countries Study, the coronary risk in Crete was only 1/15th of that in Finland, where most of the fat was saturated fat (accompanied by cholesterol), and 40% of that in Japan, where the diet is a low-fat diet favouring fish. (Reproduced by permission of the artist, Cornelia Blik, from Willet and Stampfer [5]).

**Figure 3 nutrients-11-00647-f003:**
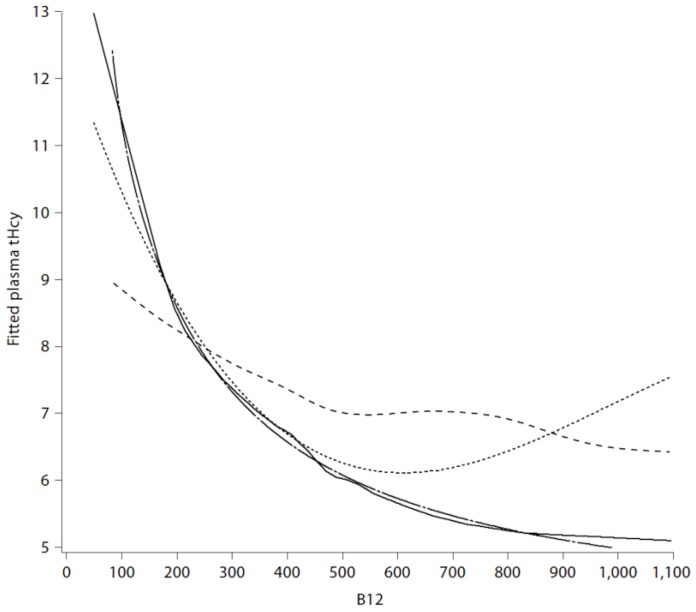
Polynomial and inverse regression, spline, loess fits for plasma tHcy with B 12. Solid, dotted, long dashed and short dashed lines represent loess, cubic polynomials, cubic polynomials of the inverse covariate and smooth splines, respectively. (Reproduced by permission of Karger from [44]).

**Figure 4 nutrients-11-00647-f004:**
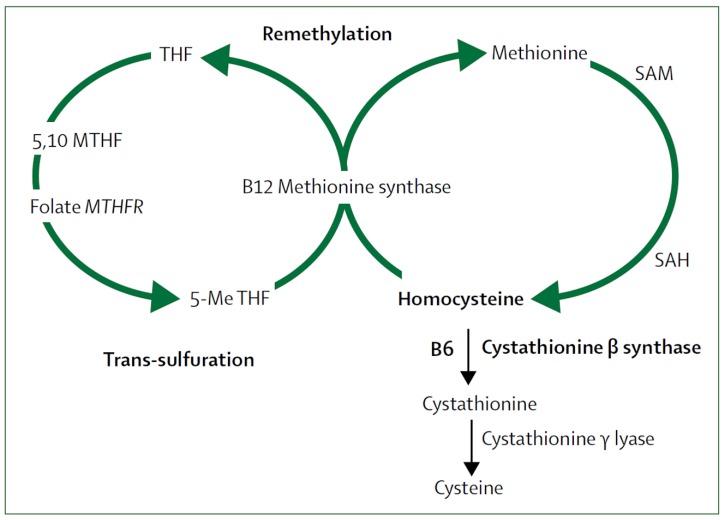
Homocysteine metabolism. B12 = cobalamin. B6 = pyridoxine. MTH = methylenetetrahydrofolate. MTHFR = methylenetetrahydrofolate reductase. SAM = S-adenosylmethionine. SAH = S-adenosylhomocysteine. 5-Me THF = 5-methyl tetrahydrofolate.

**Table 1 nutrients-11-00647-t001:** Online videos about egg industry propaganda.

http://nutritionfacts.org/video/eggs-and-cholesterol-patently-false-and-misleading-claims/
http://nutritionfacts.org/video/eggs-vs-cigarettes-in-atherosclerosis/
http://nutritionfacts.org/video/egg-cholesterol-in-the-diet/
http://nutritionfacts.org/video/how-the-egg-board-designs-misleading-studies/

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
