# Peer review of "Nutrition and Risk of Stroke"

_nutrients, 2019, doi:10.3390/nu11030647_

Reviewer 1 Report

The author highlights an important yet under recognized impact of diet on stroke risk. It is a valuable contribution. A few suggestions are noted below:

Given the focus on diet, I would encourage the author to include a section on fish and omega 3 intake and its association with stroke.

Paragraph lines 39-45 does not make sense in it's location and should be deleted or moved.

Line 56: Add “(Figure 1)”

Separate Paragraph lines 72-76 to represent it as part of Figure 2 rather than part of the written text.

Author Response

The author highlights an important yet under recognized impact of diet on stroke risk. It is a valuable contribution. A few suggestions are noted below:

Given the focus on diet, I would encourage the author to include a section on fish and omega 3 intake and its association with stroke.

A section has been added.

Paragraph lines 39-45 does not make sense in it's location and should be deleted or moved.

Perhaps the line numbers have changed. I do not see a problem in those lines.

Line 56: Add “(Figure 1)”

Figure 1 is referred to in line 46.

Separate Paragraph lines 72-76 to represent it as part of Figure 2 rather than part of the written text.

Perhaps the line numbers have changed. The caption for figure 2 is separated from the text at lines 105-107.

Reviewer 2 Report

This is an interesting narrative review on stroke and nutrition, as a neurologist reading this review I learned new content and ideas although at some moments the content is a little bit superficial and not completely clear if evidence based or not. Therefore, a table on what is in the guidelines on stroke and nutrition would be more than welcome for this review in order to know what is evidence based and what it is not and just observational data or even speculations or personal opinions.

Some of the figures and tables included in this review are the same than those include in a similar review: Spence JD. Diet for stroke prevention. Stroke Vasc Neurol. 2018;3(2):44-50. doi: 10.1136/svn-2017-000130.

Sentence 242 and 251 is repeated in the same paragraph.

Ref 58 states “The potential benefits of B vitamin therapy with folic acid and methylcobalamin or hydroxycobalamin, instead of cyanocobalamin, to lower homocysteine concentrations in people at high risk of stroke warrant further investigation.” But in this review the same author states that “we should be using that” . Is there any new evidence to support this? If yes add if not please elaborate more.

Some info on nutrition depending  on stroke subtypes or even on ICH might be interesting

References 32, 42 must be cited properly

Reference 56 is indicated as “in press” but it is already a published article

Author Response

This is an interesting narrative review on stroke and nutrition, as a neurologist reading this review I learned new content and ideas although at some moments the content is a little bit superficial and not completely clear if evidence based or not. Therefore, a table on what is in the guidelines on stroke and nutrition would be more than welcome for this review in order to know what is evidence based and what it is not and just observational data or even speculations or personal opinions.

There is very little mention of diet in the guidelines. The AHA guideline says only:“It is reasonable to counsel patients with a history of stroke or TIA to follow a Mediterranean-type diet instead of a low-fat diet.” This has been added to the text.

Some of the figures and tables included in this review are the same than those include in a similar review: Spence JD. Diet for stroke prevention. Stroke Vasc Neurol. 2018;3(2):44-50. doi: 10.1136/svn-2017-000130.

Yes; this is correct; the figures are reproduced by permission of the publishers.

Sentence 242 and 251 is repeated in the same paragraph.

I do not understand this comment, assuming that the numbers are line numbers. The sentence at line 242 says “eat egg yolks: even one egg a week (52 a year) would constitute nearly 200 extra..”, and the sentence at line 251 says “(The recipes can be downloaded from http://www.imaging.robarts.ca/SPARC/ .)”  Neither of these occurs more than once in the manuscript.

Ref 58 states “The potential benefits of B vitamin therapy with folic acid and methylcobalamin or hydroxycobalamin, instead of cyanocobalamin, to lower homocysteine concentrations in people at high risk of stroke warrant further investigation.” But in this review the same author states that “we should be using that” . Is there any new evidence to support this? If yes add if not please elaborate more.

The evidence is reviewed in the manuscript; that statement appears only in the abstract.

Some info on nutrition depending  on stroke subtypes or even on ICH might be interesting

There is little information about this; a section has been added under diet and a section about omega 3 fatty acids at the request of another reviewer; in that section there is also now mention of stroke subtypes.

References 32, 42 must be cited properly

Reference 32 appears to be correct: Bogiatzi C, Gloor G, Allen-Vercoe E, et al. Metabolic products of the intestinal microbiome and extremes of atherosclerosis. Atherosclerosis. 2018;273:91-97.  Reference 42 has been updated to Juraschek SP, Miller ER, 3rd, Weaver CM, Appel LJ. Effects of Sodium Reduction and the DASH Diet in Relation to Baseline Blood Pressure. J Am Coll Cardiol. 2017;70(23):2841-2848.

Reference 56 is indicated as “in press” but it is already a published article

This has been updated.

Round  2

Reviewer 2 Report

Comments have been addressed